# Glucocorticoid Receptor: A Multifaceted Actor in Breast Cancer

**DOI:** 10.3390/ijms22094446

**Published:** 2021-04-24

**Authors:** Lara Malik Noureddine, Olivier Trédan, Nader Hussein, Bassam Badran, Muriel Le Romancer, Coralie Poulard

**Affiliations:** 1Université de Lyon, F-69000 Lyon, France; Lara.noureddine@lyon.unicancer.fr (L.M.N.); olivier.tredan@lyon.unicancer.fr (O.T.); Muriel.LEROMANCER-CHERIFI@lyon.unicancer.fr (M.L.R.); 2Inserm U1052, Centre de Recherche en Cancérologie de Lyon, F-69000 Lyon, France; 3CNRS UMR5286, Centre de Recherche en Cancérologie de Lyon, F-69000 Lyon, France; 4Laboratory of Cancer Biology and Molecular Immunology, Faculty of Sciences, Lebanese University, Hadat-Beirut 90656, Lebanon; nader_hussein@yahoo.com (N.H.); pr.bassambadran@gmail.com (B.B.); 5Centre Leon Bérard, Oncology Department, F-69000 Lyon, France

**Keywords:** breast cancer, glucocorticoid receptor, glucocorticoids, OCDO, coregulators

## Abstract

Breast cancer (BC) is one of the most common cancers in women worldwide. Even though the role of estrogen receptor alpha (ERα) is extensively documented in the development of breast tumors, other members of the nuclear receptor family have emerged as important players. Synthetic glucocorticoids (GCs) such as dexamethasone (dex) are commonly used in BC for their antiemetic, anti-inflammatory, as well as energy and appetite stimulating properties, and to manage the side effects of chemotherapy. However, dex triggers different effects depending on the BC subtype. The glucocorticoid receptor (GR) is also an important marker in BC, as high GR expression is correlated with a poor and good prognosis in ERα-negative and ERα-positive BCs, respectively. Indeed, though it drives the expression of pro-tumorigenic genes in ERα-negative BCs and is involved in resistance to chemotherapy and metastasis formation, dex inhibits estrogen-mediated cell proliferation in ERα-positive BCs. Recently, a new natural ligand for GR called OCDO was identified. OCDO is a cholesterol metabolite with oncogenic properties, triggering mammary cell proliferation in vitro and in vivo. In this review, we summarize recent data on GR signaling and its involvement in tumoral breast tissue, via its different ligands.

## 1. Introduction

Breast cancer (BC) is the deadliest cancer among women worldwide, followed by lung and colorectal cancer. The International Agency for Research on Cancer (IARC) approximated that BC accounted for the death of over 626,679 women worldwide in 2018 and estimated the diagnosis of 2 million new cases [1]. It is predicted that one in eight women will develop BC during their life.

BC is a complex heterogeneous disease that encompasses a variety of subtypes with diverse clinical, morphological, and molecular features [2,3]. BC is molecularly classified based on the expression of common biomarkers: estrogen/progesterone receptors (ERα/PR) and human epidermal growth factor receptor-2 (HER2) [4], and is further subclassified into: Luminal A, Luminal B, HER2-enriched, and triple-negative breast cancer (TNBC) (including Basal-like and Claudin-low). Luminal A and B are the most predominant ERα/PR-positive tumors, and the Luminal B subtype is distinguished by the high expression of Ki67 (proliferation marker) and occurrence of HER2-positivity. HER2-enriched tumors are characterized by high HER2 cell surface expression. TNBCs are defined as ERα-negative, PR-negative, and HER2-negative, among which the Basal-like subtype is frequently associated with BRCA1-mutations [5,6,7]. Hence, different treatment strategies are required. Currently, surgical resection (lumpectomy, mastectomy) is the most common localized therapy for patients with non-metastatic BC. In parallel, systemic therapy is established for BC based on its subtype. ERα-positive breast tumors receive endocrine therapy to block ERα activity, including anti-estrogens such as selective ERα modulators (SERM) (i.e., tamoxifen), selective downregulators (SERD), such as fulvestrant, or estrogen synthesis inhibitors such as aromatase inhibitors. The standard endocrine treatment for premenopausal patients is 5 years of tamoxifen, or 5 years of aromatase inhibitors for postmenopausal patients. HER2/ERBB2-positive breast tumors receive intravenous medicine that specifically targets the HER2 protein, such as Trastuzumab, combined with chemotherapy. However, TNBC have no specific treatment and mainly receive cytotoxic intravenous chemotherapy [8,9].

Synthetic glucocorticoids (GCs) such as dexamethasone (dex), derived from steroidal endogenous glucocorticoids, are widely used as an adjuvant for chemotherapy in BC treatment to prevent hypersensitivity reactions through binding to its glucocorticoid receptor (GR) [10]. Recent studies demonstrated that GCs decrease estrogen-induced cell proliferation in ERα-positive BCs [11]. While other investigations stated that glucocorticoid treatment in TNBCs inhibits chemotherapy-induced cell apoptosis [12], and induces metastasis [13], thus raising new concerns.

This review summarizes the various effects of glucocorticoid receptor, and its ligands on breast tumor progression, and aims to further decipher how GR-signaling is regulated in BC.

## 2. Glucocorticoid Receptor

Human GR was initially isolated in 1985 from the BC cell line MCF-7 by the group of Pierre Chambon [14]. Human GR (h-GR) is encoded by a single gene the “nuclear receptor subfamily 3 group c member 1” (*NR3C1*) localized in the chromosome 5 short arm (5q31.3) [15]. The *NR3C1* gene is composed of 9 exons, in which exons 2-9 encode for the GR protein [16]. Exon 1 encodes for the 5′-untranslated region (5′-UTR) known as the promoter region of GR. This region has distinct features as it lacks TATA or CAT boxes and presents an extensively GC-rich motif. Moreover, this region possesses various binding sites for transcription factors (TFs) [17]. Alternative splicing and translation initiation have yielded multiple GR protein isoforms, including the classical 777 amino acid GRα and the 742 amino acid long GRβ (Figure 1A). The latter exists at a lower level compared to GRα. Both isoforms possess identical amino acids up to amino acid 727, but then differ with GRα containing 50 non-homologous AA in its C-terminus, whereas GRβ only exhibits 15 AA [18,19]. This difference at the C-terminus levels confers special features to the GRβ isoform. GRβ is neither able to bind to endogenous GCs nor to activate glucocorticoid-responsive reporter/endogenous genes and mainly resides in the cell nucleus [20]. Additionally, GRβ works as an antagonist of GRα. Indeed, several studies demonstrated its dominant-negative impact on GRα-induced transcriptional activity by competing on GR-responsive elements (GRE) and through the binding of coregulators and formation of functionally inactive GRα/GRβ heterodimers [21,22]. As the GRα isoform is responsible for most GC-mediated transcriptional activities, we will focus on GRα in this review, and will hereafter refer to it as GR.

GR is a protein ubiquitously expressed in the body [18,23,24,25]. GR belongs to the nuclear hormone receptor (NHR) superfamily and displays the common three functional domains, namely a highly conserved DNA-binding domain (DBD), the ligand-binding domain (LBD), and other regulatory N- and C-terminal domains (Figure 1A). Most of the post-translational modifications of GR occur in its N-terminal domain (Figure 1B).

## 3. GR Ligands

### 3.1. Glucocorticoids (GCs)

The natural GC cortisol is a cholesterol-derived hormone, named based on its role in maintaining glucose homeostasis. GCs are primarily synthesized and secreted by the adrenal gland cortex upon cytokine stimulation of the hypothalamic-pituitary-adrenal (HPA) axis, where the corticotropin-releasing hormone (CRH) secreted by the hypothalamus acts on the anterior-pituitary to produce adrenocorticotropic hormone (ACTH). This latter in turn triggers GC secretion by the adrenal gland. During basal and unstressed conditions, GCs are secreted in a circadian manner, however, their release is further increased due to physiological (i.e., increased immune response) and emotional stress [26,27]. Once released into the circulation, plasma proteins bind and transport inactive GCs into tissues. Most of the secreted GCs (around 90%) bind to corticosteroid-binding globulin (CBG) [28]. Their lipophilic nature allows them to diffuse passively through the plasma membrane into the cytosol. However, a balance between active and inactive forms of GCs controls the amount of GC available. Two enzymes regulate GC availability in the cytoplasm, namely the 11β-hydroxysteroid dehydrogenase-1 (11β-HSD1) that converts cortisone (inactive) to cortisol (active), and the 11β-hydroxysteroid dehydrogenase-2 (11β-HSD2) which drives the opposite reaction [29,30] (Figure 2). 

Biologically active GCs bind to GR to exert their broad physiological roles on many different cells, tissues, and organs. GCs regulate many different physiological pathways including glucose metabolism, immune response, central nervous system (cognition, mood, sleep), reproduction, cardiovascular function, development, cell death, and maintenance of vascular tone [31] (Figure 3).

### 3.2. Synthetic GCs

GCs were used for the first time in the late 1940s by Dr. Philip Hench to treat rheumatoid arthritis. Dr. Hench received the Nobel Prize in Medicine for this discovery [32]. Pharmaceutical industries have since developed various synthetic GCs, including Prednisolone, Methylprednisolone, Fluticasone, Budesonide, and Dexamethasone, used as treatments for several diseases. All of these synthetic GCs share a similar structure to that of endogenous GCs, albeit with optimized features. Indeed, they are more (i) potency; synthetic variants activate GR better than cortisol, (ii) specific; synthetic GCs such as dexamethasone (dex) exclusively bind to GR, whereas endogenous GCs can activate both GR and Mineralocorticoid Receptor, and (iii) controllable; synthetic GCs can be processed by 11β-HSD1/2 (like dex) or not (like prednisolone), thus controlling their availability [33,34].

GCs are mainly known as anti-inflammatory and immunosuppressive therapeutics, used to treat asthma, allergies, rheumatoid arthritis, multiple sclerosis, and systemic lupus erythematous. Moreover, they are used to prevent transplant rejection. Nevertheless, their success is hindered by two major drawbacks: the long-term high dose treatment induces (i) adverse side effects such as hypertension, skin atrophy, hyperglycemia, growth retardation, osteoporosis, cardiovascular diseases, and (ii) tissue-specific glucocorticoid resistance due to chronic GC treatment [34,35,36].

Besides, they have been used in clinical oncology for nearly 70 years [37]. They are routinely administered to treat hematological malignancies to foster cell apoptosis by inducing pro-apoptotic genes and inhibiting survival genes [38,39]. In non-hematological cancers, such as breast and prostate cancers, GCs are used as chemotherapy or radiotherapy adjuvants to alleviate side effects. For instance, GC treatment increases appetite, reduces fatigue, and prevents vomiting and allergic reactions [40].

### 3.3. OCDO

Recently, a cholesterol-derived oncometabolite, the 6-oxo-cholestan-3β,5α-diol (OCDO), also called cholestane-6-oxo-3,5-diol or yakkasterone (CAS N° 13027-33-3), was identified as a GR-ligand [41,42]. OCDO is the oxidative product of the carcinogenic cholestane 3β,5α,6β-triol (CT) catalyzed by 11β-hydroxysteroid-dehydrogenase-type-2 (11βHSD2), the enzyme inactivating cortisol into cortisone [42] (Figure 2). CT is generated from cholesterol-5,6-epoxides (5,6-ECs) through cholesterol-5,6-epoxide hydrolase (ChEH) [43]. OCDO was shown to promote BC cell proliferation in vitro and in vivo independently of ERα by activating the nuclear localization of GR, regulating its transcriptional activity, and consequently inducing cell cycle progression [42]. Moreover, higher levels of OCDO and its synthesizing enzymes ChEH and 11βHSD2 were detected in BC patient samples compared to normal tissues, and further mRNA database analyses indicated that the overexpression of these enzymes was correlated with a higher risk of patient death [42]. In normal breast, the concentration of OCDO was measured at 25 nM, whereas a concentration of 1 μM was reported in breast tumors [42]. The effects of OCDO can be inhibited by impairing its synthesis with ChEH inhibitors (e.g., Dendrogenin A, DDA) and 11βHSD2 silencing or by antagonizing GR with mifepristone [42,44].

## 4. GR Signaling

GR mediates its functions in cells through the binding of its ligands. Without ligand binding, the GR monomer resides predominantly in the cell cytoplasm in a resting state as a part of a multiprotein complex with chaperons and FK506 immunophilins proteins in a high ligand binding affinity conformation. This complex is also implicated in GR maturation, activation, and nuclear transport. Hormone binding triggers GR conformational change and activation, thus liberating GR from the chaperone-associated proteins and exposing its two NLS [45,46]. GR then translocates into the nucleus via its pores and binds to DNA either directly at high-affinity chromosomal sites known as GREs or indirectly through other TFs via protein-protein interactions (Figure 4). Direct GR-DNA interactions occur in multiple ways. (i) GR binds as a homodimer to glucocorticoid-binding sites (GBS) on DNA, (ii) GR binds as a monomer to inverted-repeat GBS on DNA, also known as negative GRE sites mainly accompanied with transcriptional repression, (iii) GR binds directly to GREs and physically interacts with other non-GR TFs on a neighbor DNA site in a composite manner, or (iv) GR activates transcription after physical interaction with other TFs, such as the proinflammatory TF AP-1 (Activator protein-1) and NF-κB [47,48,49,50]. Through all these mechanisms, GR was shown to regulate up to 10–20% of the human genome in different cell types [51]. 

In addition to the classical genomic ligand-dependent GR pathway, several studies have reported that unliganded GR also modulates cell signaling (Figure 4). Interestingly unliganded GR was described to display a protective role in BC, as it was shown to bind to the promoter region of a tumor suppressor gene, *BRCA1*, upregulating its expression in non-malignant mammary cells. Conversely, exposure to GCs induces a loss of GR recruitment to the BRCA1 promoter concomitant to a decrease in BRCA1 expression, highlighting the role of GCs in inducing BC [52]. Moreover, gene expression microarray analysis identified 343 target genes upregulated and 260 downregulated by unliganded GR in mammary epithelial cells. Some of the positively regulated genes were involved in pro-apoptotic signals. Moreover, unliganded GR regulated the cholesterol 25-hydroxylase (Ch25h) gene in a similar manner to BRCA1, as the association of unliganded GR to the promoter of Ch25h gene was disrupted by GCs [53]. Liganded and unliganded GR could work as a balance for controlling differentiation and apoptosis, where unliganded GR may be a mechanism for reducing BC risk by eliminating abnormal cells.

DNA-bound GR recruits coregulator complexes forming transcription regulatory complexes. These coregulators can function as corepressors or coactivators, resulting in local chromatin compaction (gene transcription repression) or local chromatin relaxation (gene transcription activation), respectively [54,55]. However, many coregulators function in both activation and repression of transcription, depending on the specific gene and cellular environment [56]. Coregulators are classified into different functional groups. The first group includes the ATP-dependent SWI-SNF chromatin-remodeling complex that catalyzes the repositioning of nucleosomes on DNA and thereby increases TF accessibility [57] where GR interacts specifically with the core subunits Brahma and BRG1 (also known as SMARCA4) of the SWI-SNF complex through its DBD, LBD and AF1 domains [58,59,60,61]. The second group consists of the histone-modifying enzymes that are responsible for adding or removing histone modifications [62,63]. These include histone methyltransferases such as PRMT4 Arginine-methyltransferase (known as CARM1) and G9a Lysine-methyltransferases (known as EHMT2) [64,65], histone acetyltransferases such as P300/CBP-PCAF and SAGA complexes [66], and histone deacetylases (HDACs) such as NCOR/SMRT-HDAC complexes [67]. However, these enzymes are able to modify other coregulators, adding another layer of complexity. Additionally, GR recruits other groups of coregulators that function as scaffold proteins responsible for recruiting other coregulators through their multiple protein-interaction domains, a well-known example is the pl60 SRC family (SRC-1, SRC-2, and SRC3), which preferentially interacts with SRC-2 (also called NCoA-2, TIF2 or GRIP1) [68].

Several hundred coregulators have been identified, indicating a high level of complexity in this process. Although each coregulator functions with multiple TFs, their actions are gene-specific, i.e., each coregulator is required only for a subset of the genes regulated by a specific TF (such as GR). These coregulator-specific gene subsets often represent selected physiological responses among multiple pathways targeted by a given transcription factor. Modulating the activity of one (or a subset of) coregulator(s) would therefore affect GC regulation of only the subset of GR target genes that requires this coregulator, thus modulating the hormone response to selectively promote or inhibit specific GC-regulated pathways [56].

Of note, GCs were described to foster non-genomic activities of GR mainly through the mitogen-activated protein kinase (MAPK) pathway in cardiovascular, immune, and neuroendocrine systems [69]. In addition, in BC, GCs increase the levels of reactive oxygen species (ROS) and reactive nitrogen species (RNS), inducing DNA damage and reducing DNA repair by dissociating GR from Src [70].

## 5. Post-Translational Modifications of GR

It is well known that the activity of proteins is tightly regulated by post-translational modifications (PTMs), which can be controlled by specific signaling pathways. To date, the function of GR is known to be affected by numerous phosphorylation events, but also by other modifications such as acetylation, ubiquitination, sumoylation, and methylation (Figure 1B). Here, we chose to focus on specific PTMs that could be relevant in BC.

### 5.1. Phosphorylation

In most cases, GR is phosphorylated at a basal level and becomes hyperphosphorylated upon ligand binding [71,72]. MAPKs, cyclin-dependent kinases, and Glycogen synthase kinase-3β (GSK-3β) are the main kinases involved in GR phosphorylation and widely implicated in BC. The specific site of GR phosphorylation determines the subsequent effect on its function. For instance, GR phosphorylated on S211 is a transcriptionally active form of the receptor [73]. Conversely, phosphorylation on S226 by c-Jun N-terminal kinase (JNK), a member of the MAPK family, was shown to abrogate GC-dependent transcriptional activity [71,74,75,76]. S404 phosphorylation by glycogen synthase kinase 3β impairs GR signaling [77]. In most cases, these phosphorylation sites alter the recruitment of major coregulators impairing GR transcriptional activity. For example, S211 phosphorylation catalyzed by p38 MAPK induces a conformational change, which facilitates coactivator recruitment (i.e., MED14) resulting in an increase in the transcriptional activity of GR [73,74]. Inversely, phosphorylation of S404 impedes GR coregulator recruitment of p300/CBP and the p65 subunit of NF-κB [77].

GR phosphorylation also modifies its localization. For example, S203 is phosphorylated by MAPK ERK1/2 in order to maintain GR in the cytoplasm and prevent its binding to the promoters of its target genes [71,76]. Furthermore, phosphorylation of GR at S134 and S226 prevents its translocation to the nucleus, impairing GC-induced gene expression [75,78].

### 5.2. Other Modifications

After ligand binding, GR is acetylated by the acetyltransferase Clock (circadian locomotor output cycles kaput) on K480, K492, K494, and K495, reducing its binding of GR to the GRE of specific target genes, impairing its transcriptional activity (Figure 1B) [79,80].

The stability of the receptor is also regulated by ubiquitinylation and sumoylation. GR is ubiquitinated at K419, targeting GR for degradation by the proteasome [81,82]. The E3 ligase CHIP (carboxy terminus of heat shock protein 70-interacting protein) was reported to be involved in this process where it modulates expression levels and activity of GR [83]. Additionally, sumoylation of GR at K277, K293, and K703, catalyzed by SUMO-1-conjugating E2 enzyme Ubc9, can regulate GR transcriptional activity on specific subsets of GR target genes [84,85]. GR sumoylation is not dependent on the ligand-binding but is rather influenced by environmental changes, potentially deregulated in BC [86].

Finally, we reported that GR is methylated by the arginine methyltransferase PRMT5 in the ERα-positive breast cancer cell line MCF-7 [87], although the targeted arginine remains to be identified.

## 6. The Role of GR in Breast Tissue

In normal breast tissue, GR is present in the nuclei and in the cytoplasm of luminal epithelial cells [88,89]. GR was also detected in the nuclei of adipocytes and of myoepithelial cells surrounding lobular and duct units. Additionally, GR is slightly expressed in the nuclei of stromal and endothelial cells. GCs were shown to be involved in the development of the mammary gland at puberty and during pregnancy [90,91]. Mechanistically, GCs stimulate the expression of β4-integrin, an extracellular protein essential for the spatial organization of the mammary epithelial acini [91].

Because GR knockout mice are not viable, authors used different approaches in order to study the role of GR in mammary gland function and development in adult mice [90,92,93]. Studies demonstrated that GR is strongly implicated in the mammary gland, though it has no effect on milk production and secretion. Cre-LoxP models in which the GR gene was specifically deleted in epithelial cells, revealed that GR is essential for cell proliferation during lobuloalveolar development [93]. Furthermore, mice lacking the DNA binding function of GR show an impairment in the ductal development of the mammary gland in virgin females, but no problem in the milk protein production. Authors suggest that DNA binding-defective GR is still able to interact with phosphorylated Stat5 proteins, involved in milk protein synthesis [92].

GCs were shown to inhibit mammary gland apoptosis during normal lactation [94]. In addition, Bertucci et al. demonstrated that GCs modulate early involution of the mammary gland. Stat5 and GR synergize to stimulate the expression of milk protein genes during lactation and act as survival factors [95]. Indeed, synthetic GCs regulate Stat5 and Stat3 signaling and inhibit apoptosis induction when administered within the first 48 h upon cessation of suckling.

## 7. The Role of GR in Breast Cancer Progression

Extensive studies have been carried out to understand the cellular and biological effects of GR on BC cell survival and progression. However, the role of GR ranges from proliferative to anti-proliferative based on ERα expression and activity (Figure 5). Indeed, GR expression has different prognostic values depending on the BC subtypes, with a high expression of GR being correlated with a worse prognosis in TNBC and with a better prognosis in early-stage ERα-positive BCs [11,96,97]. At the transcriptional level, literature converges to establish that GC drives the expression of pro-tumorigenic genes in ERα-negative BCs [98,99], but inhibits ERα transcriptional activity and E2-mediated cell proliferation in ERα-positive BCs [100,101,102].

### 7.1. ERα-Positive BCs

A high expression of GR in BCs is correlated with a better prognosis and relapse-free survival outcome in early stages for ERα-positive BC patients [11,96,97] (Figure 5A). In vitro experiments demonstrated the ability of GCs to inhibit the proliferation of ERα-positive BC MCF-7 models by altering cell cycle progression [102,104]. Mechanistically, this should occur by a direct interaction of ERα with GR, through the GR DNA-binding domain, regulating ERα transcriptional activity and therefore E2-stimulated proliferation [11,100,101,105]. Further assessments using Chromatin Immunoprecipitation (ChIP) experiments in MCF-7 cells revealed that GR displaced ERα and the coactivator SRC3 at the ERα-response elements (ERE) of specific target genes, either by direct recognition of ERE or through indirect binding with other factors such as AP-1, thus antagonizing ERα activity [101,106,107] (Figure 5B). Further studies reported that GR and ERα coactivation enhanced GR binding to GR- and ERα-responsible elements (GRE and ERE), resulting in an increase in pro-differentiating genes and negative regulators of pro-oncogenic Wnt signaling, and a decrease in mesenchymal transition related genes expression, thus improving relapse-free survival in ERα-positive BCs [11]. Moreover, a recent study demonstrated that liganded GR, regardless of the nature of the ligand (i.e., GR agonist or GR antagonist) decreased E2-mediated proliferation by suppressing the association between ERα and chromatin at the enhancer region of E2-induced pro-proliferative genes, subsequently reducing their expression [105]. GR sumoylation is also involved in this process. Indeed, Yang et al. demonstrated that GR recruitment to the ERα enhancer requires GR sumoylation on K277, K293, and K703, and subsequent recruitment of the NCor/SMRT/HDAC3 corepressor complex, repressing the estrogen (E2) program (Figure 5A). In addition, E2 treatment promotes the expression of the PP5 phosphatase, inducing the dephosphorylation of GR on S211, decreasing the activity of GR on specific GR target genes involved in cell growth arrest [108]. Further studies in T47D cells demonstrated that dex treatment inhibits cell migration by disrupting their cytoskeletal dynamic organization by impairing the AKT/mTOR/RhoA pathway [109]. However, the specific mechanisms underlying this process were not elucidated.

It is known that GR expression is repressed predominantly in ERα-positive breast tumors due to two distinct mechanisms: methylation of its promoter at CpG islands [110,111] and proteasomal degradation [112]. Interestingly, methyl-CpG islands in the GR promoter work as a binding site for Kaiso, a pox virus, and zinc finger (transcription factor), resulting in the repression of GR expression in ERα-positive breast cancer cells (MCF-7 and T47D), attenuating GR anti-apoptotic activity [113]. In addition, using an engineered MCF-7/GR cell line, Archer’s group showed that estrogen agonists, but not ERα antagonists enhance proteasomal degradation of GR via Mdm2, impacting its transcriptional activity [112]. However, as this cell line expresses 100,000 times more GR than MCF-7 cells, additional experiments will be needed to confirm this result in a more physiological context.

Altogether, these data suggest that GR mediates the repression of the transcriptional program of ERα in ERα-positive BCs via a crosstalk between GR and ERα.

### 7.2. ERα-Negative BCs

In contrast to ERα-positive breast cancer, GR expression was associated with poor outcome, shorter BC-specific survival, and earlier relapse at early-stages of human ERα-negative BCs [96,97,98]. Indeed, a retrospective meta-analysis of 1378 early stage ERα-negative BCs and 623 TNBCs confirmed that a high tumoral GR expression was significantly correlated with a shorter relapse-free survival in BC patients, whether undergoing treated or not with adjuvant chemotherapy [96,114]. Furthermore, in the last few years, a growing body of evidence clearly demonstrated the tumorigenic effects of GCs in ERα-negative BCs, as evidenced by resistance to chemotherapy and metastasis formation [12,13,98]. A genome-wide study identified specific dex-induced GR target genes, involved in tumor cell survival and chemotherapy resistance, EMT, chromatin remodeling, and epithelial cell/inflammatory cell interactions, suggesting the involvement of GR in the aggressive behavior of ERα-negative BCs [96]. Recently, global gene expression and GR ChIP-sequencing analyses identified a signature of a specific subset of GR target genes involved in cell survival, cell invasion, and chemoresistance [114] (Figure 5C).

Different mechanistic investigations are ongoing to further understand the role of GR in driving tumor progression in ERα-negative BCs. Among them, it was demonstrated that cellular stress, such as oxidative stress or hypoxia, in primary TNBCs or ERα-negative BC cell lines, increases the phosphorylation of GR on S134, thus potentiating stress signaling mediated by GR activation leading to an increase in the expression of breast tumor kinase BRK, known as protein tyrosine kinase 6 (PTK6), essential for aggressive BC phenotypes [115]. In addition, TNBCs express high levels of functionally active pS134-GR in comparison to luminal BCs, which could explain why GR expression is correlated with a better prognosis in luminal BCs than in TNBCs [116]. Recently, research on patients and TNBC cell line-derived xenograft models, revealed that GR activation at distant metastatic sites, due to an increase in GC levels, promotes BC colonization and reduces the overall survival by upregulating the expression of ROR-1 kinase, a receptor tyrosine kinase-like orphan receptor-1, previously shown to be implicated in BC [13,117,118]. Indeed, downregulation of ROR-1 by shRNAs decreases metastasis and prolongs survival in mouse models. These studies support previously published expression microarray analyses that identified several kinases as promising targets for in ERα-negative BC treatment [119,120].

Additional investigations linked BC progression and chemoresistance to the disruption of the oncosuppressor Hippo pathway, which is mainly composed of kinase complexes, transcriptional cofactor Yes associated-protein (YAP) and its paralog WW domain-containing transcription regulator 1 (TAZ), and TEA domain transcription factors (TEAD1-4). The high expression and activity of YAP/TEAD-4 was reported to contribute to BC cell survival and progression [121]. Recent studies demonstrated that GR activation by dex dysregulated the Hippo pathway by inducing the transcriptional activity, nuclear accumulation, and protein/RNA levels of YAP and TEAD-4. Functionally, this activation of YAP and TEAD-4 led to cell survival, metastasis, chemo-resistance, and cancer stem cell self-renewal in vitro and in vivo [99,122]. TEAD-4 along with its coactivator, the Krüppel-like factor 5 (KLF5), a pro-survival TF, were among the nine genes reported to be overexpressed in high-grade ERα-negative tumors [123] and their high expression level was associated with poor prognosis and shorter survival in BC patients [122,124]. Moreover, it was shown that TEAD-4 forms a complex with KLF5 and promotes TNBC cell proliferation by inhibiting p27 gene transcription [125]. Interestingly, GR activation by dex upregulates KLF5 expression in TNBCs, and high KLF5, in turn, induces cisplatin resistance in vitro and in vivo [126].

Global gene expression analyses in MDA-MB-231 cells revealed that several pro-survival genes were induced by dex treatment (i.e., SGK1 (Serum and glucocorticoid-regulated kinase-1), MKP-1 (MAPK phosphatase-1)) [127,128,129]. Additionally, ChIP-seq analyses on the same cell line revealed that dex-liganded GR binds to GREs of pro-tumorigenic genes driving drug resistance and TNBC progression [99]. The transcriptional activation of these pro-survival genes by GR upon dex treatment contributes to inhibiting paclitaxel or doxorubicin-induced apoptosis in MDA-MB-231 cells [98,128]. Conversely, the degradation of GR and disruption of its anti-apoptotic signaling using the Hsp90 inhibitor was shown to enhance TNBC sensitivity to paclitaxel in vitro and in vivo [130].

Furthermore, in vivo studies were carried out to investigate the potential inhibitory effect of GCs on anti-tumoral paclitaxel activity. Accordingly, the pre-treatment with dex significantly attenuated the therapeutic efficacy of paclitaxel on human tumor xenografts established from transplanting human ERα-negative BCs into nude mice [126,130,131,132]. However, the pre-treatment of TNBCs with the GR antagonist Mifepristone in parallel to dex and Paclitaxel potentiated the cytotoxic efficacy of the chemotherapy, by inducing caspase-3/PARP cleavage-mediated cell death and blocking GR-mediated survival signaling by antagonizing GR-induced SGK1 and MKP1 gene expression. In addition, it was reported that mifepristone pre-treatment decreased MDA-MB-231 xenograft tumor growth [12]. Consistent with these observations, a randomized Phase I clinical trial showed that GR is a promising target in TNBCs, as patients with GR-positive and triple-negative tumors responded to the combination of GR antagonism (mifepristone) and paclitaxel [133].

Lately, investigators reported that GR is essential for EMT and metastasis induction in BCs. They found that high GR expression levels suppress the transcription of insulin receptor substrate 1 (IRS-1), which is a cytoplasmic adaptor protein that transmits insulin/insulin-like growth factor signals. IRS-1 suppression by GR activates extracellular regulated protein kinase 2 (ERK2) and induces EMT [134]. Moreover, they showed that in the absence of GR ligands, GR is transcriptionally activated in TNBCs through its phosphorylation on S134 by p38, following the homeostatic sensing of intrinsic stress or extrinsic factors (like TGFβ1). Phospho-S134-GR activates the p38 MAPK stress-signaling pathway, leading to TNBC cell anchorage-independent growth and migration [116].

## 8. Concluding Remarks

This review underlines the implication of GR and its ligands in BC biology and physiology. The fact that GR expression has different prognostic values depending on the BC subtypes, highlights an unanticipated level of complexity. The repression of the ERα transcriptional program in ERα-positive BCs is known to be linked to a crosstalk between GR and ERα. However, a growing body of evidence clearly demonstrates the tumorigenic effects of GR in ERα-negative BCs, as evidenced by resistance to chemotherapy and metastasis formation. In addition, proliferative effects of OCDO are GR-dependent regardless of the hormonal status of the BC. However, the transcriptional program of OCDO in the different subtypes of BCs has so far not yet been identified and could provide clues to its oncogenic properties. In addition, because OCDO is a cholesterol-derived oncometabolite, a more global analysis of the expression of enzymes producing OCDO is of utmost importance following the status of BCs, in addition to the cholesterolemia status of patients, to fully understand the impact of OCDO on breast tumorigenesis in ERα-positive vs. ERα-negative BCs.

In BC treatment, synthetic GCs are commonly used for their antiemetic, anti-inflammatory, and energy and appetite-stimulating properties, and thus help to manage the side effects of chemotherapy. However, in the last few years, increasing evidence clearly shows the tumorigenic effects of GR in ERα-negative BCs, including resistance to chemotherapy and metastasis formation. Targeting GR activity is not an option because of its pleiotropic activity. However, GCs are often the only option for patients to counteract the effects of chemotherapy. Because coregulator-specific gene subsets are often unique to selected physiological responses among the multiple pathways regulated by a TFs [58], modulating the activity of one (or a subset of) coregulator(s) could therefore affect GC regulation of only selected GR target genes requiring this coregulator, and may enable the modulation of the hormonal response to selectively promote or inhibit specific GC-regulated pathways. To illustrate this concept, we demonstrated that coactivator activity of the GR coregulator G9a is modulated by methylation/phosphorylation, which regulates distinct physiological pathways, including migration of the lung cancer cell line A549 [135] and GC-induced cell death in leukemia [136,137].

Of interest in this field, in a cohort of BC patients, a Danish epidemiological study reported no impact of GC use on BC recurrence, irrespective of the route of administration or combined chemotherapy [138]. Because GCs are prescribed to counteract the side effect of chemotherapy depending on the level of discomfort, the doses of GCs received cannot be fully monitored. Additional epidemiological studies will be interesting to confirm these observations in different patient cohorts. Moreover, a growing body of evidence suggests the impact of stressful events on BC risk. Indeed, the Women Health Initiative Study showed that an acute stress event can be associated with increased BC risk [139]. Rats stressed by chronic social isolation present higher levels of corticosterone, associated with a dysregulated GR distribution. Among socially-isolated animals, GR was more often found in the nucleus compared to the cytoplasm in tumor samples, and these rats harbored more aggressive mammary tumors [140].

Even though this review mainly presents the effect of GCs on tumor cells, we cannot exclude that they also affect the tumor microenvironment, particularly cancer-associated fibroblasts [141,142]. GCs were shown to regulate the proliferation of myofibroblasts and have major roles in wound healing [143]. Moreover, Catteau et al. found that GR is expressed in 73% of CAFs in BC [144], associated with tumoral grade or Ki67 expression. Taking in account GR expression in the tumor environment following the classification of tumor status, could be utmost importance and may serve as an interesting target in the regulation of the tumoral breast microenvironment.

Emerging data are highlighting the importance of a second form of estrogen receptor, ER beta (ERβ), in breast cancer biology (for review [145]). As a study performed in the central nucleus of amygdala showed that ERβ activation prevents glucocorticoid-induced anxiety behaviors and reduced cortisol levels in the plasma of rats compared to animals implanted with vehicle or GR agonist [146], further studies will be needed to investigate the potential cross-talk between ERβ and GR in BC.

Despite incredible breakthroughs in our understanding of BC, and the key role of GR in the pathology, major challenges in this field of research still remain.

## Figures and Tables

**Figure 1 ijms-22-04446-f001:**
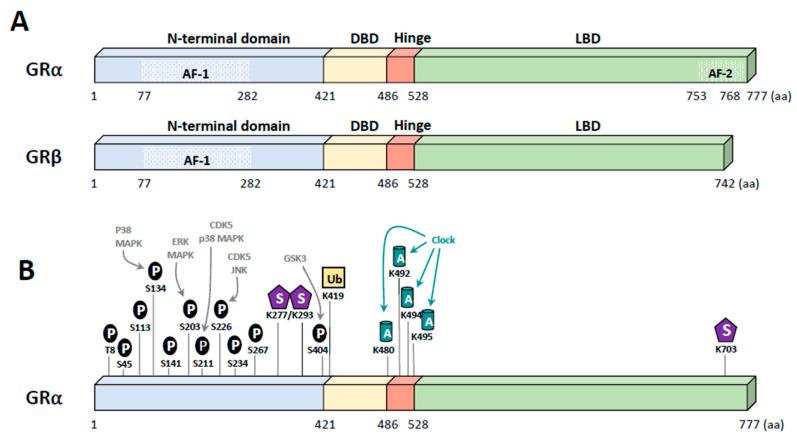
Structure of human glucocorticoid receptor (GR). (**A**) Alternative splicing of exon 9 results in two isoforms of GR; GRα and GRβ. GR contains different domains: the N-terminal domain (NTD), DNA binding domain (DBD), the flexible Hinge region and the ligand binding domain (LBD). GR encompasses two activation functions (AF-1 and AF-2) allowing the recruitment of coregulators and the transcriptional machinery. (**B**) GRα undergoes numerous post-translational modifications including phosphorylation of various residues (mainly serine residues) (P), sumoylation (S), acetylation (A) and ubiquitinylation (Ub).

**Figure 2 ijms-22-04446-f002:**
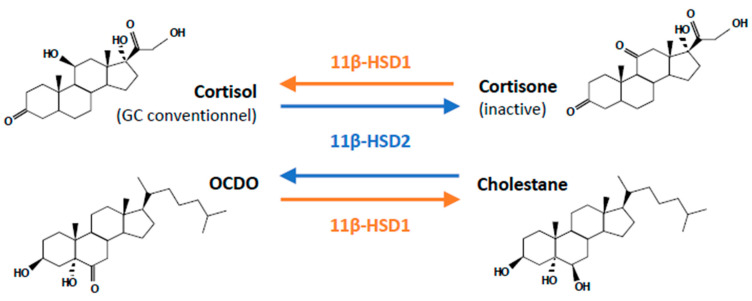
Structure of natural GR ligands. The natural form of glucocorticoid is cortisol that can be converted into inactive cortisone by the 11β-HSD2 enzyme. The same enzyme metabolizes cholestane-3β,5α,6β-triol (CT) into 6-oxo-cholestan-3β,5α-diol (OCDO).

**Figure 3 ijms-22-04446-f003:**
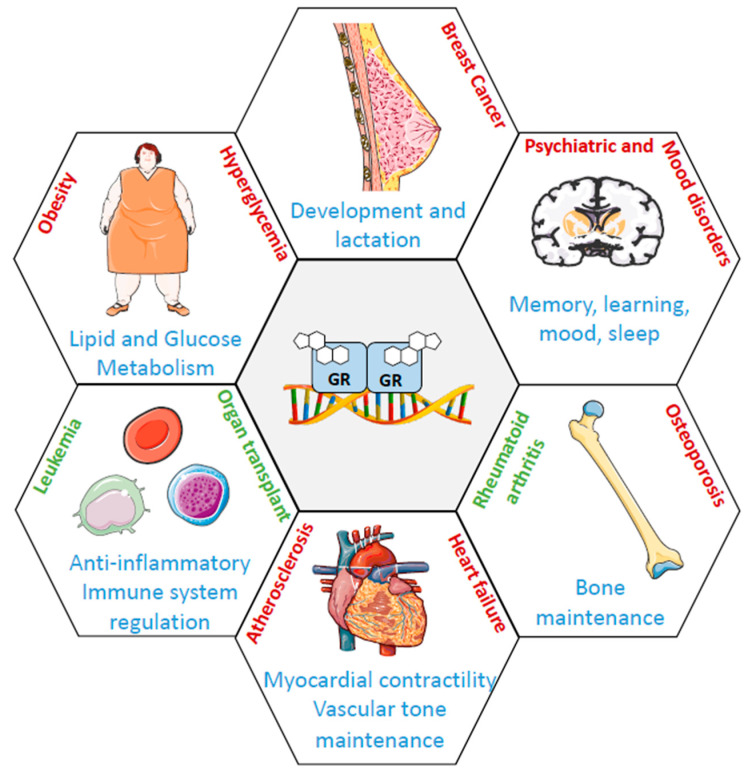
GR involvement in human health and disease. The schematic diagram represents the roles of GR in major systems (blue label) with beneficial roles of synthetic GCs used in clinics (green label) and adverse effects of GCs (red label).

**Figure 4 ijms-22-04446-f004:**
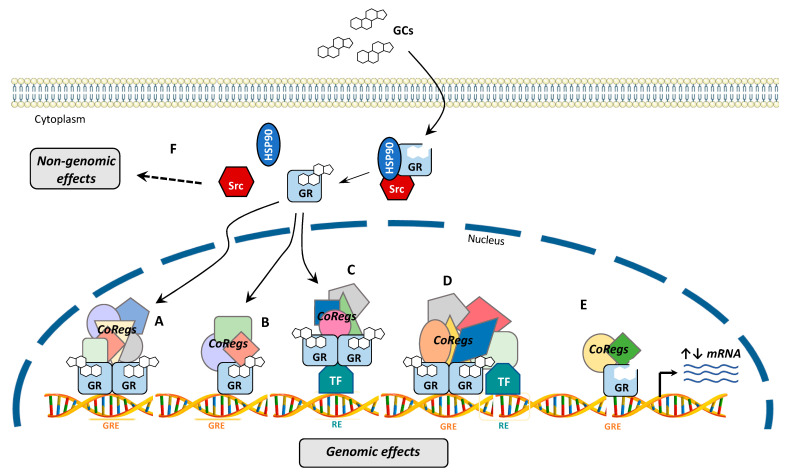
Genomic GR signaling. Upon ligand binding, GR undergoes a conformational change, dissociates from chaperone proteins (HSP90), and translocates into the nucleus, where it can bind directly to DNA as a dimer on a specific GR response element (GRE) (**A**), as a monomer through a simple GRE (**B**), through other transcription factors (TFs) by tethering itself to the TF (**C**), or in a composite manner by directly binding to GRE (**D**). Unliganded GR modulates cell signaling in the absence of GCs (**E**). In addition to the genomic action of GR in the nucleus (**A**–**E**), when GR dissociates from its cytoplasmic complex upon GCs treatment, it can also regulate non-genomic effects (**F**). Specific sets of coregulators are recruited, resulting in the activation or repression of target genes, regulating specific biological functions.

**Figure 5 ijms-22-04446-f005:**
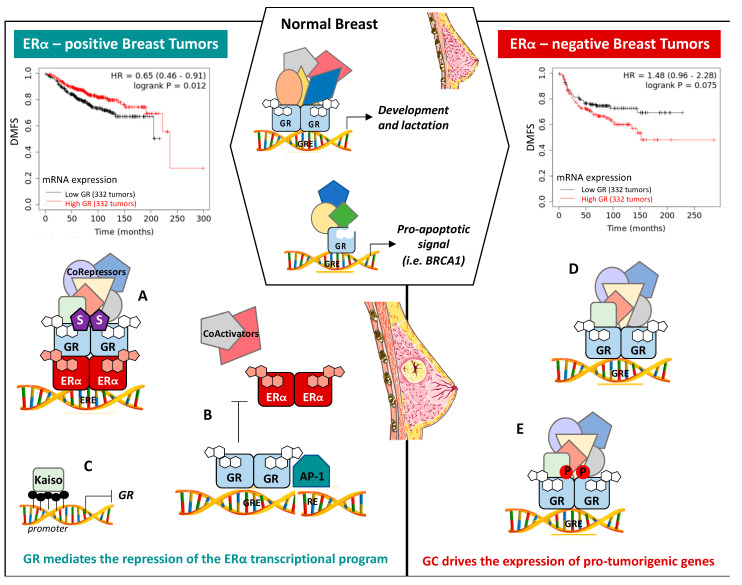
GR involvement in physiological and pathophysiological breast functions. Mammary gland development and lactation are regulated by GR in normal breasts. In addition, unliganded GR binds to the promoter region of some pro-apoptotic genes, such as *BRCA1*, upregulating their expression in non-malignant mammary cells. GR controls the outcome of BC depending on the ERα status of the tumor. In ERα-positive BCs, GR regulates the repression of the ERα transcriptional program by directly binding to ERα, promoting its sumoylation (S) and recruitment of corepressors (**A**), or in a composite manner by directly binding to AP-1 (**B**). In addition, in ERα-positive breast cancer cells, methyl-CpG islands in the GR promoter work as a binding site for Kaiso, resulting in the repression of GR expression (**C**). Conversely, in ERα-negative BCs, GR regulates pro-tumorigenic genes and is associated with a worse prognosis (**D**), and pS134-GR is found to be higher in TNBC in comparison with luminal BCs and associated with a migratory phenotype (**E**). Kaplan–Meier curves were built using KM plotter database [103].

## Data Availability

Not applicable.

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
