# Peer review of "Glucocorticoid Receptor: A Multifaceted Actor in Breast Cancer"

_ijms, 2021, doi:10.3390/ijms22094446_

Round 1
Reviewer 1 Report
While generally a very good review of the field, for a review intended to review the role of GR in breast cancer it does have a lot of basic and not immediately relevant material at the beginning. It also does not touch on a number of key results relevant to the role of GR in breast cancer. There is no mention of the unliganded activity of GR, and while the role of this aspect of GR regulation is relatively underappreciated in the field it has been shown to have relevance to breast cancer in particular. It could also be illustrated in Figure 4. The role of methylation of the GR promoter in ER+ breast cancer was not mentioned and is a major omission. While the pro-oncogenic aspect of the the use glucocorticoids is referred to, there is significant epidemiological evidence to suggest this is not an issue. A more balanced approach to the evidence in this area would be appropriate.
The role of GR in the maintenance of lactation and post-lactational regression was also not mentioned at all, and given its relationship to breast cancer at many levels this should be addressed..
It should also be clear that permission to use the Kaplan Meier curves in Figure 5 was obtained from Conzen. Also Figure 5C was not as informative as expected from the text.
The role of psychological stress and its link to GR and breast cancer was also not touched on in this review, but may be outside of the purview of this type of review.
Reviewer 2 Report
The present study by Noureddine et. al. holds great importance in terms of breast cancer (BC) management, knowing the fact that many chemotherapies use Glucocorticoid as an adjuvant besides its adverse effects on ER-negative BC. However, there are certain aspects of the presented topic that needs to be addressed.
Major points:
- In lines 344 to 347, the authors discuss that irrespective of the nature of the ligand present, the activation of the GR receptor in ER-positive BC has a better survival outcome. Does that mean that high expression of OCDO in ER-positive BC indicates the same? The authors mention that OCDO is higher in ER-positive BC as compared to normal BC.
As the studies suggest that OCDO unlike other glucocorticoids does not interact with ER-alpha and mediates its oncogenic effect directly via binding to GR (PMID: 23953592). Furthermore, it has been shown that it has a completely different mechanism than other GR agonists or antagonists (PMID: 30120211).
Therefore, it is essential to re-evaluate the author's findings in light of these factors and discuss them in detail.
- Authors have discussed that oxidative stress or hypoxia can lead to phosphorylation of GR at S134 (lines 377-382) in TNBC or ER-negative BC. How would the authors explain why this is not the case in ER-positive tumors as ER is known to induce oxidative DNA damage in BC (PMID: 14514655).
- As most of the ER expression analysis is done for the presence of ER-alpha but not ER-beta, it would be important to discuss whether the GR receptor has a similar mechanism of action for both ER alpha and ER beta or it has a contradictory role. Recent studies indicate that TNBC does express high levels of ER-beta (PMID: 30338035).
- Furthermore, the expression of GR is also well studied in the stromal cells of BC especially fibroblasts. It would be important to discuss the role of GR in the tumor microenvironment in respect of ER-positive and ER-negative BC.
- A study showed that estrogen receptor leads to GR degradation via MDM-2 protein in presence of ER agonist (PMID: 12897156). It would be essential to discuss this as hormonal therapy is given in hormonal receptor-positive BC (ER-positive), so if ER degrades GR then how does the expression of high GR in ER-positive BC can be possible and could be responsible for better survival outcomes.
Minor comments:
- In lines 42-43, the authors mention that luminal B and luminal A could be differentiated based on Ki67 expression. Of not luminal B could also be HER2-positive/negative.
- In lines 70-71, the authors stated ‘The h-GR gene is composed of 9 exons, in which exons 2-9 en-70 code for the GR protein’, I suppose it is ‘NR3C1’ not ‘h-GR’ gene.
- In lines 42-43, authors stated ‘TNBCs are defined as ER-negative, PR-negative and 42 HER2-negative, with the Basal-like subtype being characterized by an enrichment in 43 BRCA1-mutations’.
Not all TNBC are BRCA1 mutated and therefore can not be characterized by the BRCA1 mutation.
Round 2
Reviewer 2 Report
The suggested changes has been incorporated in the manuscript. It is a well written paper and the manuscript could be accepted in the present form.